# Network Robustness as a Mathematical Property: training, evaluation and attack

## Abstract

Neural networks are widely used to detect general patterns in noisy data, however they are also known to not be particularly robust, i.e. moving a small distance in the input space can change the network's output significantly. Recently, many methods for improving network robustness have been proposed and this growing body of research has given rise to numerous explicit or implicit notions of robustness. Connections between these notions are often subtle, and a systematic comparison of these different definitions is missing in the literature.

In this paper we attempt to address this gap by performing an in-depth comparison of the different definitions of robustness by analysing their relationships, assumptions, interpretability and verifiability. By viewing robustness as a stand-alone mathematical property, we are able to show that, having a choice of several definitions of robustness, one can combine them in a modular way when defining training modes, evaluation metrics, and attacks on neural networks. We also perform experiments to compare the applicability and efficacy of different training methods for ensuring the network obeys these different definitions.

## 1 Introduction

Safety and security are critical for some complex AI systems involving neural networks, yet they are difficult to ensure. The most famous instance of this problem is guaranteeing robustness against *adversarial attacks* (Szegedy et al., 2014; Goodfellow et al., 2015). Intuitively, an image is $\epsilon$-*ball robust* if, when you move no more than $\epsilon$ away from it in the input space, the output does not change much, or alternatively, the classification decision that the network gives does not change. *Adversarial robustness* is a property that even very accurate neural networks fail to satisfy.

The proposed solution is to (re)train the network with robustness specifically in mind. Such training can be seen as a way to impose a formal specification, and so may contribute to explainability as well as verification. This work considers four of the most prominent families of techniques:

1. *Data augmentation* (Shorten & Khoshgoftaar, 2019)
2. *Adversarial training* (Goodfellow et al., 2015; Madry et al., 2018)
3. *Lipschitz robustness training* (Anil et al., 2019; Pauli et al., 2021)
4. *Training with logical constraints* (Xu et al., 2018; Fischer et al., 2019)

The last technique, *training with logical constraints*, is a more general approach that can train for not just robustness, but a wide range of constraints expressed in some logical language.

Although the first three families of methods all seek to represent the same high-level knowledge in the neural network, each technique seeks to optimise for subtly different definitions of robustness. We formally identify these as *standard* (SR), *classification* (CR), *Lipschitz* (LR) and *strong classification* (SCR) robustness. Given these differences, some natural questions to ask are: What are the relationships between them? What assumptions do they make about the training dataset? Are some more effective than others? Are some more interpretable by users than others?

**Contributions.** In this work, we attempt to answer these questions. We take classification problems as an example domain, interpreting a neural network $f : \mathbb{R}^n \to \mathbb{R}^m$ as a procedure that separates

the $n$-dimensional data into $m$ classes. This enables the following findings: We observe that from the security perspective, different definitions of robustness ultimately determine the nature of attack, thus giving rise to SR, CR, LR, SCR attacks. Thus, one can train for example with SCR constraint but attack with SR constraint. This raises questions about relative strengths of these different training methods and attacks. Some constraint-driven training methods are special cases of others. For example, adversarial training known in the literature can be seen as a form of training with SR constraint, with certain amount of parameter tuning.

We can order some robustness constraints based on their strength, for example, we show that LR *implies* SR, and SCR *implies* CR. In this case, training with a stronger constraint (e.g. LR) will protect better against both kinds of attacks (in this case, both SR and LR attacks). Some pairs of constraints cannot be ordered by strength (e.g. SR and SCR, LR and SCR), and in this case, optimising training for a given constraint defends better only against attacks with this same target constraint. Moreover, we show that training with logical constraints defends against adversarial attacks better than data augmentation for any choice of robustness definition as a training constraint. Finally, we show that there are additional common criteria that can be used to qualitatively compare different modes of constraint-driven training, e.g. interpretability, global or local nature. E.g. CR is the most interpretable, but not globally desirable, LR is least interpretable, but globally desirable.

To our knowledge, this is the first systematic study of robustness training from the point of view of the impact of precise formalisation of robustness on training, evaluation and attack. Note that some of the previous work reported on unstable performance of constraint-driven training when defending against attacks (Ayers et al., 2020), which we do not observe in our experiments. Some papers (Fischer et al., 2019) listed and even implemented some kinds of robustness constraints that we study here, but gave no indication of their relative performance. We are not aware of any prior analysis of SR, LR, CR, SCR abstractly as logical constraints.

The paper is organised as follows. Section 2 explains how different robustness constraints arise from different machine learning approaches to constraint-driven training. Section 3 abstractly analysises these robustness definitions, establishing their relative strength, interpretability and applicability. Section 4 shows how these robustness constraints determine different evaluation metrics and attacks, and provides a comprehensive empirical evaluation of the robustness constraints deployed as training constraints and as attacks. Section 5 concludes the paper and outlines future work.

## 2 EXISTING TRAINING TECHNIQUES

**Data augmentation** is one of the simplest methods of improving the robustness of a neural network via training (Shorten & Khoshgoftaar, 2019). It is applicable to any transformation of the input (e.g. addition of noise, translation, rotation, scaling) that leaves the output label unchanged. To make the network robust against such a transformation, one augments the dataset with instances sampled via the transformation. Although it may seem that this simple solution has nothing to do with formal logic, it imposes significant choices from the point of view of the constraint specification:

**c1.** the choice of $\epsilon$ will reflect our assumptions about admissible range of perturbations;

**c2.** the choice of the distance function $|\cdot - \cdot|$ that measures the $\epsilon$-proximity will reflect our assumptions on desirable geometric properties of the perturbations;

**c3.** the choice of the sampling method (random sampling, adversarial attacks, generative algorithm, prior knowledge of images etc.) will determine the constraint we optimise for.

But, perhaps even more significantly for us, this method determines the exact definition of robustness that we optimise for when we train our neural network $f : \mathbb{R}^n \to \mathbb{R}^m$. We call it *classification robustness* and formally define as follows: given a training dataset input-output pair $(\hat{\mathbf{x}}, \mathbf{y})$ and a distance function $|\cdot - \cdot|$, for all inputs $\mathbf{x}$ within the $\epsilon$-ball distance of $\hat{\mathbf{x}}$, ensure that class $y$ has the largest score in output $f(\mathbf{x})$. In other words:

**Definition 1 (Classification robustness)**
$$CR(\epsilon, \hat{\mathbf{x}}) \triangleq \forall \mathbf{x} : |\mathbf{x} - \hat{\mathbf{x}}| \leq \epsilon \Rightarrow \arg\max f(\mathbf{x}) = \mathbf{y}$$

Used as a spec for training, this constraint does not account for possibility of having "uncertain" images in the dataset, for which a small perturbation ideally should change the class. For datasets

that contain a significant number of such images, it will lead to significant reduction in accuracy of the trained neural networks; and, as we show later it may even have a detrimental effect on a network's robustness.

**Adversarial training** is the current state-of the-art method to robustify a neural network. Whereas standard training tries to minimise loss between the predicted value, $f(\hat{\mathbf{x}})$, and the true value, $\mathbf{y}$, for each entry $(\hat{\mathbf{x}}, \mathbf{y})$ in the training dataset, adversarial training minimises the loss with respect to the worst-case perturbation of each sample in the training dataset. It therefore replaces the standard training objective $\mathcal{L}(\hat{\mathbf{x}}, \mathbf{y})$ with:

$$\max_{\forall \mathbf{x}: |\mathbf{x} - \hat{\mathbf{x}}| \leq \epsilon} \mathcal{L}(\mathbf{x}, \mathbf{y}) \tag{1}$$

Algorithmic solutions to the maximisation problem that find the worst-case perturbation has been the subject of several papers. The earliest suggestion was the Fast Gradient Sign Method (FGSM) algorithm introduced by Goodfellow et al. (2015):

$$\text{FGSM}(\hat{\mathbf{x}}) = \text{proj}(\hat{\mathbf{x}} + \epsilon \cdot \text{sign}(\nabla_{\mathbf{x}} \mathcal{L}(\mathbf{x}, \mathbf{y})))$$

However, modern adversarial training methods usual rely on some variant of the Projected Gradient Descent (PGD) algorithm (Gu & Rigazio, 2014) which iterates FGSM some number of times:

$$\text{PGD}_0(\hat{\mathbf{x}}) = \hat{\mathbf{x}}; \quad \text{PGD}_{t+1}(\hat{\mathbf{x}}) = \text{PGD}_t(\text{FGSM}(\hat{\mathbf{x}}))$$

It has been empirically observed that neural networks trained using this family of methods exhibit greater robustness at the expense of an increased generalization error (Tsipras et al., 2018; Madry et al., 2018; Zhang et al., 2019), which is frequently referred to as the *accuracy-robustness tradeoff* for neural networks (although this effect has been observed to disappear as the size of the training dataset grows (Raghunathan et al., 2019).

In logical terms what is this procedure trying to train for? Obviously it's unreasonable to expect that adversarial training will ever succeed in driving the loss of all perturbations down to zero. Therefore let us assume that there's some maximum distance, $\delta$, that it is acceptable for the output to be perturbed given the size of perturbations in the input. This leads us to the following definition, where $|| \cdot - \cdot ||$ is a suitable distance function over the output space:

**Definition 2 (Standard robustness)**

$$SR(\epsilon, \delta, \hat{\mathbf{x}}) \triangleq \forall \mathbf{x} : |\mathbf{x} - \hat{\mathbf{x}}| \leq \epsilon \Rightarrow ||f(\mathbf{x}) - \mathbf{y}|| \leq \delta$$

In the case of adversarial training the distance between the outputs $||\mathbf{x} - \mathbf{y}||$ is equal to $\mathcal{L}(\mathbf{x}, \mathbf{y})$.

We note that, just as with data augmentation, choices **c1 – c3** are still there to be made, although the sampling methods are usually given by special-purpose FGSM/PGD heuristics based on computing the loss function gradients.

**Training for Lipschitz robustness.** More recently, a third competing definition of robustness has been proposed: Lipschitz robustness (Balan et al., 2018). Inspired by the well-established concept of Lipschitz continuity, Lipschitz robustness asserts that the distance between the original output and the perturbed output is at most a constant $L$ times the change in the distance between the inputs.

**Definition 3 (Lipschitz robustness)**

$$LR(\epsilon, L, \hat{\mathbf{x}}) \triangleq \forall \mathbf{x} : |\mathbf{x} - \hat{\mathbf{x}}| \leq \epsilon \Rightarrow ||f(\mathbf{x}) - \mathbf{y}|| \leq L|\mathbf{x} - \hat{\mathbf{x}}|$$

As will be discussed in Section 3, this is a stronger requirement than standard robustness. Techniques for training for Lipschitz robustness include formulating it as a semi-definite programming optimisation problem (Pauli et al., 2021) or including a projection step that restricts the weight matrices to those with suitable Lipschitz constants (Gouk et al., 2021).

**Training with logical constraints.** Logically, this discussion leads one to ask whether a more general approach to constraint formulation may exist, and several attempts in the literature addressed this research question (Xu et al., 2018; Fischer et al., 2019), by proposing methods that can translate a first-order logical formula $C$ into a *constraint loss function* $\mathcal{L}_C$. The loss function penalises the

Table 1: *A comparison of the different types of robustness studied in this paper.*

| Definition | Standard robustness | Lipschitz robustness | Classification robustness | Strong classification robustness |
|---|---|---|---|---|
| Symbol | $SR(\epsilon, \delta)$ | $LR(\epsilon, L)$ | $CR(\epsilon)$ | $SCR(\epsilon, \eta)$ |
| Problem domain | General | General | Classification | Classification |
| Interpretability | Medium | Low | High | Medium |
| Globally desirable | ✓ | ✓ | ✗ | ✗ |
| Has loss functions | ✓ | ✓ | ✗ | ✓ |
| Adversarial training | ✓ | ✗ | ✗ | ✗ |
| Data augmentation | ✗ | ✗ | ✓ | ✗ |
| Logical-constraint training | ✓ | ✓ | ✗ | ✓ |

network when outputs do not satisfy a given Boolean constraint, and universal quantification is handled by a choice of sampling method. Our standard loss function $\mathcal{L}$ is substituted with:

$$\mathcal{L}^*(\hat{\mathbf{x}}, \mathbf{y}) = \alpha \mathcal{L}(\hat{\mathbf{x}}, \mathbf{y}) + \beta \mathcal{L}_C(\hat{\mathbf{x}}, \mathbf{y}) \tag{2}$$

where weights $\alpha$ and $\beta$ control the balance between the standard loss and the constraint loss.

This method looks deceivingly as a generalisation on the previous approaches. However, even given suitable choices for **c1 – c3**, classification robustness cannot be modelled via a constraint loss in the DL2 framework, as $argmax$ is not differentiable. Instead Fischer et al. (2019) define an alternative constraint we will call *strong classification robustness*:

**Definition 4 (Strong classification robustness)**

$$SCR(\epsilon, \eta, \hat{\mathbf{x}}) \triangleq \forall \mathbf{x} : |\mathbf{x} - \hat{\mathbf{x}}| \leq \epsilon \Rightarrow f(\mathbf{x})_c \geq \eta$$

which looks only at the prediction of the true class and checks whether it is greater than some value $\eta$ (chosen to be 0.52 in their work).

In summary, we have hopefully demonstrated how non-trivial knowledge representation choices and problems arise on the boundary between logical form of the desired constraints and their machine-learning realisations as loss functions. In the next section, we analyse the advantages and disadvantages of each definition in order to help people better make these choices in future.

## 3 COMPARISON OF DEFINITIONS

Table 1 shows a summary of the points discussed in this section. The first aspect we discuss is the logical relationship between the various definitions, i.e. when the definitions agree and disagree.

**Standard and Lipschitz robustness.** The easiest relationship to quantify is the one between standard robustness and Lipschitz robustness. In particular, the latter is a strictly stronger constraint than the former, in the sense that when a network satisfies $LR(\epsilon, L)$ then it also satisfies $SR(\epsilon, \epsilon L)$. However, the converse does not hold, as standard robustness does not relate the distances between the inputs and the outputs. Consequently, there are $SR(\epsilon, \delta)$ robust models that are not $LR(\epsilon, L)$ robust for any $L$, as for any fixed $L$ one can always make the distance $|\mathbf{x} - \hat{\mathbf{x}}|$ arbitrarily small in order to violate the Lipschitz inequality.

**(Strong) classification robustness.** The next relationship we discuss is that of classification robustness and strong classification robustness. As discussed earlier, the latter is designed to over-approximate the former whilst providing a logical loss function with a meaningful gradient. We work under the assumption that the last layer of the classification network is a softmax layer and therefore the output forms a probability distribution. When $\eta > 0.5$ then, as would be hoped, any network that satisfies $SCR(\epsilon, \eta)$ also satisfies $CR(\epsilon)$. For $\eta \leq 0.5$ this relationship breaks down as the true class may be assigned a probability greater than $\eta$ but may still not be the class with the highest probability. We therefore recommended that one only uses value of $\eta > 0.5$ when using strong classification robustness (for example $\eta = 0.52$ in Fischer et al. (2019)).

Given that LR is stronger than SR and SCR is stronger than CR, the obvious question is whether there is a relationship between these two groups? In short, the answer to this question is no. In particular, although the two sets of definitions agree whether a network is robust around images with high-confidence, they disagree over whether a network is robust around images with low confidence. We illustrate this with an example, comparing SR against CR. We note that similar analysis holds for any pairing from the two groups.

**Standard vs classification robustness.** The key insight is that standard robustness bounds the drop in confidence that a neural network can exhibit after a perturbation, whereas classification robustness does not. Figure 1 shows two hypothetical images from the MNIST dataset. Our network predicts that Figure 1a has an 85% chance of being a 7. Now consider adding a small perturbation to the image and consider two different scenarios. In the first scenario the output of the network for class 7 decreases from 85% to 83% and therefore the classification stays in the same. In the second scenario the the output of the network for class 7 decreases from 85% to 45%, and results in the classification changing from 7 to 9. When considering the two definitions, a small change in the output leads to no change in the classification and a large change in the output leads to a change in classification and so robustness and classification robustness both agree with each other.

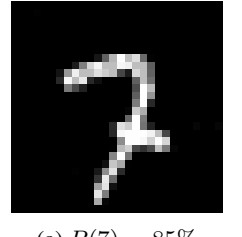 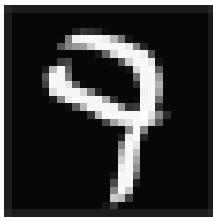

(a) $P(7) = 85\%$      (b) $P(7) = 51\%$

Figure 1: *Hypothetical images from the MNIST dataset*

However now consider Figure 1b with relatively high uncertainty. In this case the network is (correctly) less sure about the image, only narrowly deciding that it's a 7. Again consider adding a small perturbation. In the first scenario the prediction of the network changes dramatically with the probability of it being a 7 increasing from 51% to 91% but leaves the classification unchanged as 7. In the second scenario the output of the network only changes very slightly, decreasing from 51% to 49% flipping the classification from 7 to 9. Now, the definitions of SR and CR disagree. In the first case, adding a small amount of noise has erroneously massively increased the network's confidence and therefore the SR definition correctly identifies that this is a problem. In contrast CR has no problem with this massive increase in confidence as the chosen output class remains unchanged. Given this, it is clear that although standard and classification robustness agree on low-uncertainty examples, classification robustness breaks down and gives what we argue are both false positives and false negatives when considering examples with high-uncertainty.

**Dataset assumptions.** A related question that we have found to be rarely discussed in the literature is what assumptions the different definitions are making about the distribution of the training data with respect to the data manifold of the true distribution of inputs.

For SR and LR it is, at minimum, desirable for the network to be robust over the entire data manifold. In the most domains the shape of the manifold is unknown and therefore it is necessary to approximate it by taking the union of the balls around the inputs in the training dataset. We are not particularly interested about whether the network is robust in regions of the input space that lie off the data manifold, but there is no problem if the network is robust in these regions. Therefore these definitions make no assumptions about the distribution of the training dataset.

This is in contrast to (S)CR. As discussed in the previous section, rather than requiring that there is only a small change in the output, they require that there is no change to the classification. This is only a desirable constraint when the region being considered does not contain a decision boundary. Consequently when one is training for some form of classification robustness, one is implicitly making the assumption that the training data points lie away from any decision boundaries within the manifold. In practice, most datasets for classification problems assign a single label instead of an entire probability distribution to each input point, and so this assumption is usually valid. However, we feel it is important to note that classification robustness is not an appropriate definition to train for if the dataset contains input points that may lie close to the decision boundaries and in such a case may result in a logically inconsistent specification.

**Interpretability.** One of the key selling points of training with logical constraints is that, by ensuring that the network obeys understandable constraints, it improves the explainability of the neural network. Each of the robustness constraints encode that "small changes to the input only result in small changes to the output", but the interpretability of each definition is also important.

All of the definitions share the relatively interpretable $\epsilon$ parameter, which measures how large a perturbation from the input is acceptable. Despite the other drawbacks discussed so far, CR is inherently the most interpretable as it has no second parameter. In contrast, SR and SCR require extra parameters, $\delta$ and $\eta$ respectively, which measure the allowable deviation in the output. Their addition make these models less interpretable.

Finally we argue that, although LR is the most desirable constraint, it is also the least interpretable. Its second parameter $L$ measures the allowable change in the output as a proportion of the allowable change in the input. It therefore requires one to not only have an interpretation of distance for both the input and output spaces, but to be able to relate them. In most domains, this relationship simply doesn't exist. Consider the MNIST dataset, both the commonly used notion of pixel-wise distance used in the input set, although crude, and the distance between the output distributions are both interpretable. However, the relationship between them is not. For example, what does allowing the distance between the output probability distributions being no more than twice the distance between the images actually mean? This therefore highlights a common trade-off between complexity of the constraint and its interpretability.

Another interpretability consideration that appears when training with logical constraints is the semantics of the parameters $\alpha$ and $\beta$ which allow the users to decide how to weight the accuracy of the network vs the importance that it adheres to the constraint. Section 1 describes how the loss function used in logical-constraint training is split into two parts: $\mathcal{L}^*(\hat{\mathbf{x}}, \mathbf{y}) = \alpha \mathcal{L}(\hat{\mathbf{x}}, \mathbf{y}) + \beta \mathcal{L}_C(\hat{\mathbf{x}}, \mathbf{y})$, where the first part is designed to maximise the accuracy of the predictions and the second encodes the constraint. However, in practice the constraints (and therefore the derived loss functions) refer to the true label $\mathbf{y}$ rather than the current output of the network $f(\hat{\mathbf{x}})$, e.g. $\forall \mathbf{x} : |\mathbf{x} - \hat{\mathbf{x}}| \leq \epsilon \Rightarrow |f(\mathbf{x}) - \mathbf{y}| \leq \delta$. This leads to scenarios where a network that *is* robust around $\hat{\mathbf{x}}$ but gives the wrong prediction, being penalised by $\mathcal{L}_C$ which on paper is designed to maximise robustness. Essentially $\mathcal{L}_C$ is trying to maximise both accuracy and constraint adherence concurrently. Instead, we argue that to preserve the intended semantics of $\alpha$ and $\beta$ it is important to instead compare against the current output of the network e.g. $\forall \mathbf{x} : |\mathbf{x} - \hat{\mathbf{x}}| \leq \epsilon \Rightarrow |f(\mathbf{x}) - f(\hat{\mathbf{x}})| \leq \delta$. Of course, this doesn't work for SCR because in order to derive the most popular class from the output $f(\hat{\mathbf{x}})$ you need $\arg \max$, the very function SCR seeks to avoid using. This is another argument why (S)CR should be avoided if possible.

## 4 EXPERIMENTS

**Evaluation metrics.** Given a particular definition of robustness, a natural question is how to quantify how close a given network is to satisfying it? We argue that there are three different measures that one should be interested in:

1. Does the constraint hold? This is a binary measure and the answer is either true or false.

2. If the constraint doesn't hold, how easy is it for an attacker to find a violation?

3. If the constraint doesn't hold, how often does the average user encounter a violation?

Based off of these measures, we define three concrete metrics: *constraint satisfaction*, *constraint security*, *constraint accuracy*[1]. Let $\mathcal{X}$ be the training dataset, $\mathbb{B}(\hat{\mathbf{x}}, \epsilon) \triangleq \{\mathbf{x} \in \mathbb{R}^n \mid |\mathbf{x} - \hat{\mathbf{x}}| \leq \epsilon\}$ be the $\epsilon$-ball around $\hat{\mathbf{x}}$ and $P$ be the right hand side of the implication in each of the definitions of robustness. Let $\mathbb{I}_\phi$ be the standard indicator function which is 1 if constraint $\phi(\mathbf{x})$ holds and 0 otherwise. The *constraint satisfaction* metric measures the proportion of the training dataset for which the constraint holds. Unfortunately, depending on the network architecture and the constraint in question, the indicator function may not always be feasible to evaluate.

---

[1]Our naming scheme differs from Fischer et al. (2019) who use the term *constraint accuracy* (without explicitly defining it in the paper) to refer to what we term *constraint security*. In our opinion, the term *constraint accuracy* is less appropriate here than the name *constraint security* given the use of an adversarial attack.

**Definition 5 (Constraint satisfaction)**

$$\mathrm{CS}(\mathcal{X}) = \frac{1}{|\mathcal{X}|} \sum_{\hat{\mathbf{x}} \in \mathcal{X}} \mathbb{I}_{\forall \mathbf{x} \in \mathbb{B}(\hat{\mathbf{x}}, \epsilon) : P(\mathbf{x})}$$

In contrast, *constraint security* measures the proportion of inputs in the training dataset such that an attack $A$ is unable to find an adversarial example for constraint $P$. In our experiments we use the PGD attack for $A$, although in general any strong attack can be used.

**Definition 6 (Constraint security)**

$$\mathrm{CR}(\mathcal{X}) = \frac{1}{|\mathcal{X}|} \sum_{\hat{\mathbf{x}} \in \mathcal{X}} \mathbb{I}_P(A(\hat{\mathbf{x}}))$$

Finally *constraint accuracy* estimates the probability of a random user coming across a counter-example to the constraint, usually referred as *1 - success rate* in the robustness literature. Let $S(\hat{\mathbf{x}}, n)$ be a set of $n$ elements randomly uniformly sampled from $\mathbb{B}(\hat{\mathbf{x}}, \epsilon)$. Then constraint accuracy follows:

**Definition 7 (Constraint accuracy)**

$$\mathrm{CL}(\mathcal{X}) = \frac{1}{|\mathcal{X}|} \sum_{\hat{\mathbf{x}} \in \mathcal{X}} \left( \frac{1}{n} \sum_{\mathbf{x} \in S(\hat{\mathbf{x}}, n)} \mathbb{I}_P(\mathbf{x}) \right)$$

Note that there is no relationship between constraint accuracy and constraint security: an attacker may succeed in finding an adversarial example where random sampling fails and vice-versa. Also note the role of sampling in this discussion and compare it to the discussion of the choice **c3** in Section 2. Firstly, sampling procedures affect both training and evaluation of networks. But at the same time, their choice is orthogonal to choosing the verification constraint for which we optimise or evaluate. For example, we measure constraint security with respect to the PGD attack, and this determines the way we sample; but having made that choice still leaves us to decide which constraint, SCR, SR, LR, or other we will be measuring as we sample.

**Choosing an evaluation metric.** It is important to note that for all three evaluation metrics, one still has to make a choice for constraint $P$, namely SR, SCR or LR, as defined in Section 2. As constraint security always uses PGD to find input perturbations, the choice of SR, SCR and LR effectively amounts to us making a judgement of what an adversarial perturbation consists of: is it a class change as defined by SCR, or is it a violation of the more nuanced metrics defined by SR and LR? Therefore we will evaluate constraint security on the *SR/SCR/LR constraints* using a *PGD attack*. This will be further analysed in the experiment **E1** below.

For large search spaces in $n$ dimensions, random sampling deployed in constraint accuracy fails to find the trickier adversarial examples, and usually has deceivingly high performance: we found $100\%$ and $> 98\%$ constraint accuracy for SR and SCR, respectively. We will therefore not discuss these experiments in detail, but refer the reader to Appendix A.

Constraint satisfaction is different from constraint security and accuracy, in that it must evaluate constraints over infinite domains rather than merely sampling from them. Verifiers such as Marabou or ERAN are built with this purpose in mind. It is out of scope for this paper to conduct experiments by this measure, and so we leave it for future work. Our preliminary experiments with Marabou confirm the generally accepted assumption that it is very hard for training techniques to achieve a high-score for constraint satisfaction.

**Networks.** We use the *FASHION MNIST* (or just *FASHION*) (Xiao et al., 2017) and the *GT-SRB* (Stallkamp et al., 2011) datasets. For our baseline architecture, we use two fully connected layers: the first layer uses the ReLU activation function, and the second uses the clamp function to restrict each output to the range $[-100, 100]$. We use a clamp function to evaluate the constraints instead of the traditional softmax function because the former is compatible with the constraint verification tools such as Marabou whereas the latter is not. The predicted classification is then taken as the output with the maximum score. However, only during training, we pass the output from the clamp function through a softmax layer before feeding it to the loss. For instance, $f_{\mathrm{MNIST}} = F_0 \circ F_1$, where $F_1 : \mathbb{R}^{784} \to \mathbb{R}^{100}$ and $F_0 : \mathbb{R}^{100} \to \mathbb{R}^{10}$, $\alpha_0 = ReLU$ and $\alpha_1(x) = \mathrm{clamp}(x, -100, 100)$.

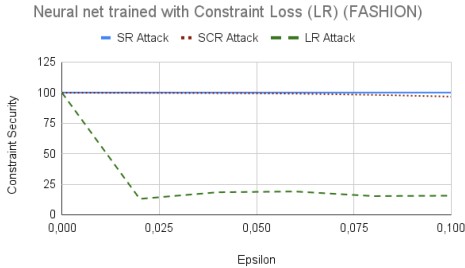
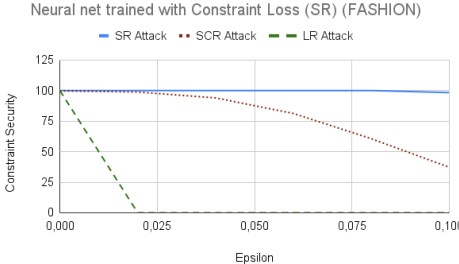

Figure 2: *Experiments that show how the two networks trained with LR and SR constraints perform when evaluated against different definitions of robustness underlying the attack.*

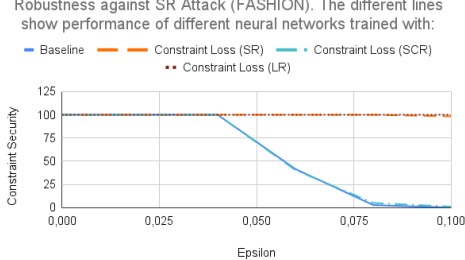
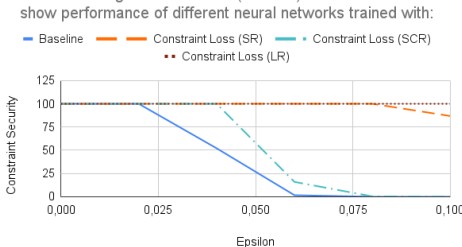

Figure 3: *Experiments that show how different choices of a constraint loss affect standard robustness of neural networks, for PGD attacks with various $\epsilon$ values.*

**Loss functions.** Since our experiments study classification problems, we will use the cross-entropy loss function as our baseline loss function:

$$\mathcal{L}_{ce}(\mathbf{x}, \mathbf{y}) = -\sum_{i=1}^{m} \mathbf{y}_i \, \log(f(\mathbf{x})_i)$$

For the $\mathcal{L}_C$ component of the loss function $\mathcal{L}^*$, we use the constraint-to-loss function translation of Fischer et al. (2019). In all experiments, we use the Adam optimiser (Kingma & Ba, 2015) with the following learning parameters: $\eta = 0.0001$, 100 epochs, a batch size of 128.

**Settings.** We keep the architecture the same throughout the experiments, and only vary the training. Thus our *Baseline* network is trained just with cross-entropy. Data Augmentation adds 2 additional images in the $0.1\text{-}\epsilon$-ball around each image, and it samples either randomly from a uniform distribution (RU) or using an FGSM attack. Adversarial training refers the training procedure described in Section 2, with FGSM sampling. In all other cases, we use a constraint loss function $\mathcal{L}_C$ defined as in Fischer et al. (2019), and we use the constraints SR, SCR, LR as defined in Section 2 with $\alpha = 1$, $\beta = 0.2$, $\epsilon = 0.1$, $\delta = 10$, $\eta = 0.52$, $L = 10$ and $L^\infty$ distance metrics. All networks trained with $\mathcal{L}^*$ use sampling by the PGD attack, for efficiency (as well as comparability).

**Results.** We start with noting the standard test set accuracy of the resulting neural networks in Table 2, making sure that our different training regimes do not deteriorate networks' general performance too drastically. The most notable accuracy drop occurs for adversarial training.

We now highlight groups of experiments that confirm or extend our main theoretical conclusions; the complete experiment description is available in Appendix A.

**Experiment set E1. Comparable constraints.** In Section 3, we established that LR as a constraint is stronger than SR, and both are not strictly comparable to SCR. This would suggest that, if we train two neural networks, one with the SR, and the other with the LR constraint, then the latter

Table 2: *Standard test set accuracy (as % of the dataset instances) for chosen trained networks.*

| Training Regime: | FASHION | GTSRB |
|---|---|---|
| Baseline | 88.2 | 92.4 |
| Data Augmentation (RU) | 88.6 | 92.8 |
| Data Augmentation (FGSM) | 88.8 | 94.5 |
| Adversarial Training | 85.1 | 83.5 |
| Constraint Loss (SR) | 88.2 | 93.3 |
| Constraint Loss (SCR) | 88.1 | 91.9 |
| Constraint Loss (LR) | 86.6 | 93.1 |

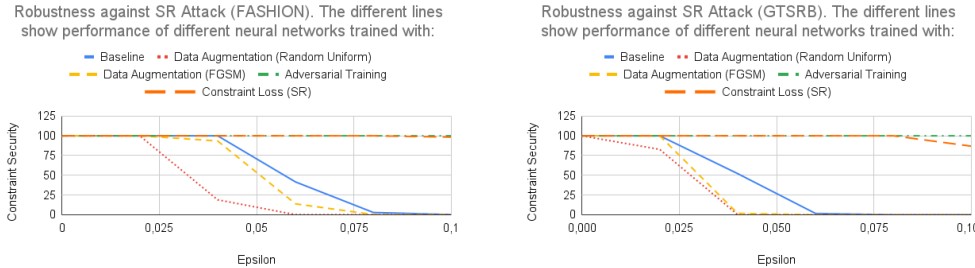

Figure 4: *Experiments that show how adversarial training, training with data augmentation, and training with constraint loss affect standard robustness of neural networks, for PGD attacks with various $\epsilon$ values.*

should always have higher constraint security against both SR and LR attacks than the former. This is indeed confirmed by the experiment shown in Figure 2. We also discussed that depending on the properties of the dataset, SR may not guarantee SCR, and Figure 2 shows exactly that case. It also confirms that generally, stronger constraints are harder to obtain: whether a network is trained with SR or LR constraints, it is less robust against LR attack than any other attack.

**Experiment set E2. Incomparable constraints.** The results in Figure 3 tell us that using the SCR constraint for training does not help to increase defences against SR attacks. A similar picture, but in reverse, can be seen when we optimise for SR but attack with SCR, see Appendix A.

**Experiment set E3. Constraint training versus data augmentation and adversarial training.** Next, we confirm our assumptions about the relative inefficiency of using data augmentation compared to adversarial training or training with constraints, see Figure 4. We show this only for SR attack, but graphs for SL and SCR attacks show the same trends, see Appendix A. Surprisingly, neural networks trained with data augmentation give worse results than even the baseline network. It is encouraging to see that Constraint Loss (SR) is almost as good as adversarial training while the latter also gives 11% drop in terms of the networks accuracy (Table 2).

**Experiment set E4. Role of other parameters.** As previously discussed, random uniform sampling struggles to find adversarial inputs in large searching spaces. It is logical to expect that using random uniform sampling when training will be less successful than training with sampling that uses FGSM or PGD as heuristics. Indeed, Figure 4 shows this effect for data augmentation, but similar trends are expected for any form of training.

Finally, one may ask whether the trends just described would be replicated for more complex architectures of neural networks. In particular, data augmentation is known to require larger networks. By replicating the results of Figure 4 with a large 18 layer convolutional network of Fischer et al. (2019) indeed confirms that larger networks handle data augmentation better, and data augmentation improves robustness compared to the baseline. Nevertheless, data augmentation still lags behind all other modes of constraint driven training, and thus this major trend remains stable across network architectures, see Appendix A.

## 5 CONCLUSIONS AND RELATED WORK

We have presented a comprehensive study of *constraint*-driven training from the formal point of view. Taking robustness as a representative constraint, we abstractly studied different forms of robustness; and showed how the existing literature on constraint-driven training can be understood through this prism. Moreover, we proposed a method that separates out the logical study of constraints from their implementation as loss functions for training on the one hand, and their use as evaluation methods and attacks on the other. We showed that this method allows us to make general conclusions about relations of different modes of constraint-driven training that were not possible before.

For translation of constraints into loss functions, we used the implementation from Fischer et al. (2019); thus our results are compatible and comparable with that prior study. In particular we have identified that the method of Fischer et al. (2019) is not a strict generalisation of the other techniques and that significant trade-offs and decisions have to be made in order to represent some of the robustness definitions.

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

## A APPENDIX

This appendix will present all the results that, due to space, were not possible to include in the main text. We will show in detail how we systematically evaluated all the trends that we reported. Figures 5, 8 are the same as Figures 4, 3, and Figures 13A, 14A correspond to Figure 2, we report them again here for completeness.

### A.1 DATA AUGMENTATION VS ADVERSARIAL TRAINING VS TRAINING WITH CONSTRAINT LOSS

Figures 5, 6 and 7 show how the networks trained with the different methods are robust against attacks. For SR and SCR attacks, Adversarial Training improves significantly the robustness of the model while training with Constraint Loss (SR) also improves it, although not as much. Training with both the Data Augmentation methods actually reduce robustness of the network. None of the training techniques succeed in ensuring robustness against LR attacks, indicating that as discussed in Section 3.1, LR is a strong property to hold. Nonetheless we report the graphs for completeness.

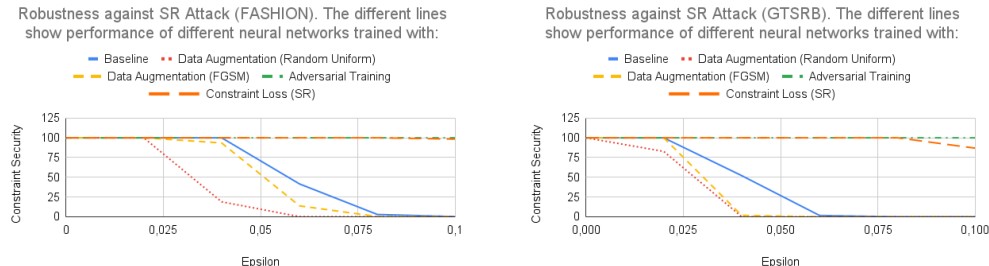

Figure 5: *Experiments that show how adversarial training, training with data augmentation, and training with constraint loss affect standard robustness of neural networks, for varying sizes of the PGD attack (measured by $\epsilon$ values).*

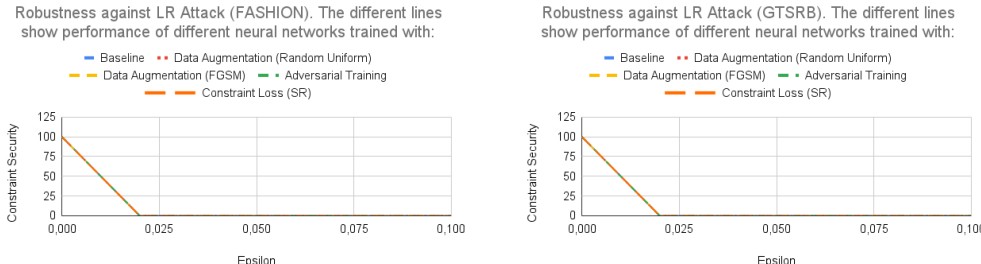

Figure 6: *Experiments that show how adversarial training, training with data augmentation, and training with constraint loss affect lipschitz robustness of neural networks, for varying sizes of the PGD attack (measured by $\epsilon$ values).*

### A.2 TRAINING WITH DIFFERENT CONSTRAINT LOSSES

Figures 8, 9 and 10 show how the networks trained with the different constraint losses are robust against attacks. For SR and SCR attacks, we can see a trend in the networks trained with the same constraint that are generally more robust against the respective attacks. On the other hand, models trained with LR are generally more robust and they have the best average improvement against all attacks. None of the training techniques succeed in ensuring robustness against LR attacks, except the network trained with LR on the FASHION dataset.

### A.3 TRAINING A BIGGER NETWORK

Figure 11 show some of the training methods scale with the architecture. On the left we have the small architecture used for all the experiments, while on the right we have a bigger architecture. The

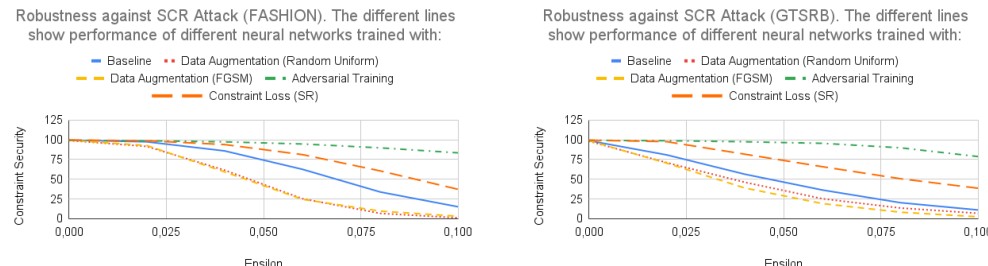

Figure 7: *Experiments that show how adversarial training, training with data augmentation, and training with constraint loss affect strong classification robustness of neural networks, for varying sizes of the PGD attack (measured by $\epsilon$ values).*

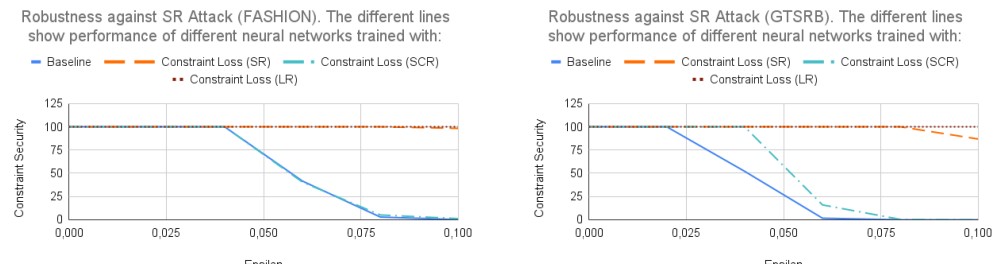

Figure 8: *Experiments that show how different choices of a constraint loss affect standard robustness of neural networks, for varying sizes of the PGD attack (measured by $\epsilon$ values).*

relative behaviour between the training methods remain the same, except from the baseline network that present a decrease in robustness and it exhibits similar behaviour to the models trained with Data Augmentation.

## A.4 NETWORKS' BEHAVIOURS AGAINST DIFFERENT ATTACKS

Previous experiments show that the most robust models are the ones trained with Adversarial Training, LR and SR. Figures 12, 13 and 14 select the relevant data from previous experiments to provide a comparison of how these models perform against the different attacks. All the networks struggle the most against LR attacks, while they present significant robustness against SR attacks and a slightly less but still important robustness against SCR attacks. Overall we can see that Adversarial Training provides the best defence against SR and SCR attacks, immediately followed by training with Constraint Loss LR. However, as already reported above, the only network that show some robustness against LR attacks is the one trained with Constraint Loss LR.

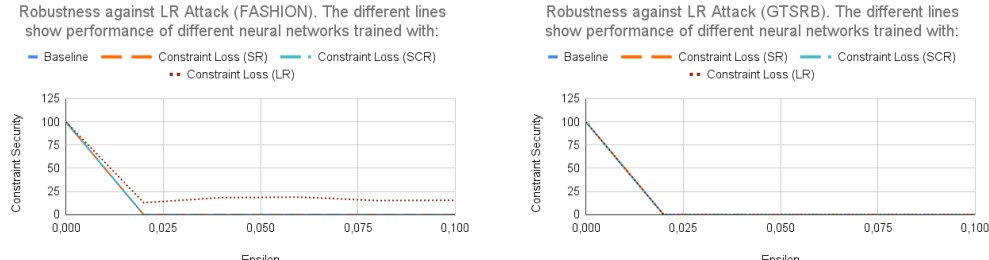

Figure 9: *Experiments that show how different choices of a constraint loss affect lipschitz robustness of neural networks, for varying sizes of the PGD attack (measured by $\epsilon$ values).*

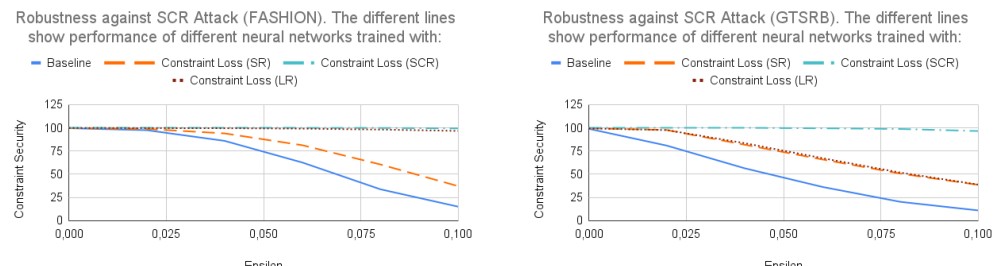

Figure 10: *Experiments that show how different choices of a constraint loss affect strong classification robustness of neural networks, for varying sizes of the PGD attack (measured by $\epsilon$ values).*

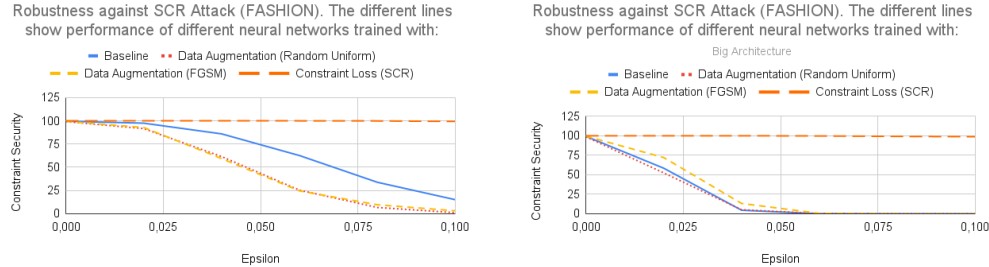

Figure 11: *Experiments that show a comparison of how training with data augmentation, and training with constraint loss affect strong classification robustness of neural networks, for varying sizes of the PGD attack (measured by $\epsilon$ values) of our standard network with a bigger architecture.*

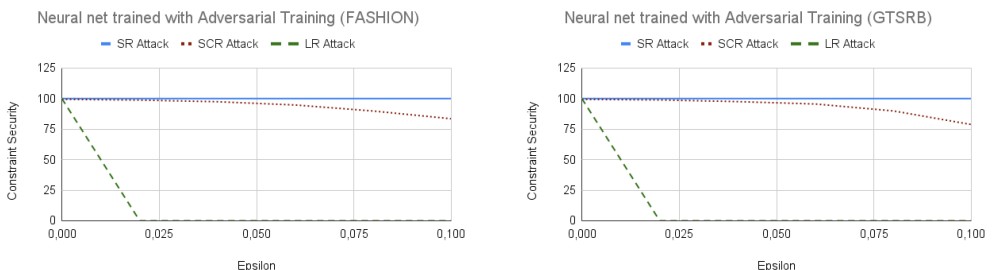

Figure 12: *Experiments that show how the networks trained with Adversarial Training perform when evaluated against different definitions of robustness underlying the attack.*

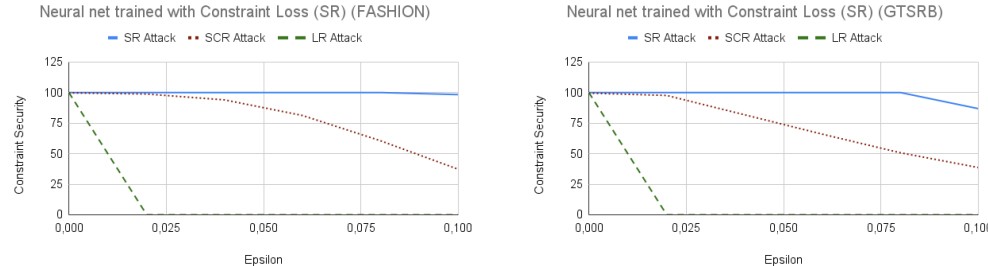

Figure 13: *Experiments that show how the networks trained with SR constraints perform when evaluated against different definitions of robustness underlying the attack.*

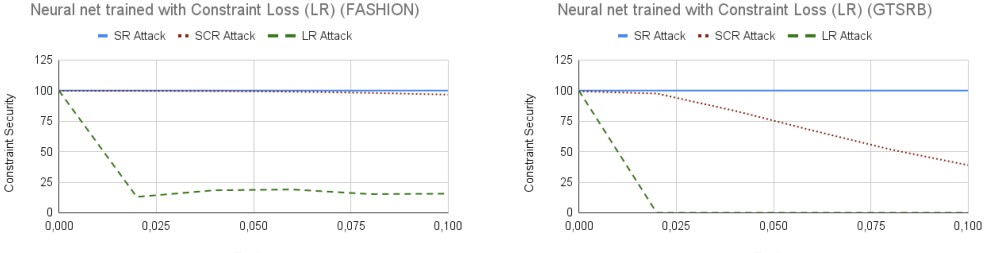

Figure 14: *Experiments that show how the networks trained with LR constraints perform when evaluated against different definitions of robustness underlying the attack.*

