# OpenReview forum: "Network robustness as a mathematical property: training, evaluation and attack"
_ICLR.cc/2022/Conference — ICLR 2022 Submitted_

### Official Review · Reviewer_2Rrw · 2021-10-18

**Correctness:** 4
**Technical Novelty And Significance:** 1
**Empirical Novelty And Significance:** 2
**Recommendation:** 3
**Confidence:** 3

**Main Review:**

Strengths:
- The summarization and proposed formal framework is very helpful in terms of unifying notions and categorizing existing work in the literature.
- Some empirical findings are interesting like LR constraint training maintains competitive robustness as adversarial training.
- Code is provided for reproducibility.

Weaknesses:
- The theoretical analyses seem a bit preliminary. The implication relationships: LR -> SR, SCR -> CR immediately follow from their definitions. The comparison table in Table 1 is mainly based on empirical reasoning.
- The empirical evaluation does not provide sufficient novel findings. It is intuitive that adversarial training provides the highest CR or SCR. Also, it is a bit straightforward that incorporating corresponding constraints provides the highest robustness against corresponding robustness notions. The implication relationships directly guarantee the findings of Experiment set E1.
- Also, the empirical evaluation lacks generalizability in terms of datasets and covered works. In terms of datasets: large-scale datasets like CIFAR-10, CIFAR-100, ImageNet, etc need to be included. In terms of covered works: the paper seems to focus heavily on the DL2 work by Fischer et al. Some work, especially for LR (see below), is not yet included. Therefore, the evaluation is not comprehensive enough.

Suggestions:
- In terms of Lipschitz robustness, recent literature imposes this constraint via proposing specialized network architectures [1,2] or Lipschitz-aware training [3,4]. These methods may need to be evaluated for comprehensiveness.
- Details on the PGD attack may need to be provided. For example, how does PGD find violations of LR? What are the used step size and iterations for PGD?

Minor:
- last paragraph of Section 1: analysises -> analyzes

[1] Trockman, Asher, and J. Zico Kolter. "Orthogonalizing Convolutional Layers with the Cayley Transform." International Conference on Learning Representations 2021.

[2] Zhang, Bohang, et al. "Towards Certifying L-infinity Robustness using Neural Networks with L-inf-dist Neurons." International Conference on Machine Learning. PMLR, 2021.

[3] Tsuzuku, Yusuke, Issei Sato, and Masashi Sugiyama. "Lipschitz-Margin Training: Scalable Certification of Perturbation Invariance for Deep Neural Networks." NIPS 2018.

[4] Leino, Klas, Zifan Wang, and Matt Fredrikson. "Globally-Robust Neural Networks." International Conference on Machine Learning. PMLR, 2021.

**Summary Of The Paper:**

This paper formally summarizes common robustness notions in the literature as: standard robustness (SR), classification robustness (CR), Lipschitz robustness (LR), and strong classification robustness (SCR). The paper proves LR implies SR, SCR implies CR. Then some empirical studies are conducted.

**Summary Of The Review:**

The summarization and formalization of robustness notions is a good contribution of this paper, which helps the community to sort out different approaches and their applicability. However, from the empirical front and theoretical front, the paper may not provide enough novel findings nor useful approaches. Thus, the paper may be below the acceptance threshold at ICLR.

---

> ### Author Response · Authors · 2021-11-22
> **We agree with the reviewer**
>
> We agree with the weaknesses that the reviewer stated and we will try to address them in future versions.

---

### Official Review · Reviewer_6o25 · 2021-11-01

**Correctness:** 2
**Technical Novelty And Significance:** 1
**Empirical Novelty And Significance:** 1
**Recommendation:** 1
**Confidence:** 4

**Main Review:**

The authors aim at creating a general and formal framework for uniting the different notions of robustness of neural networks. In particular, they consider four notions.

First, the restriction to only four approaches in such a vast field of research is not preferable for a general framework (just as an example, there are probabilistic robustness [1], stability training [2], targeted robustness [3]).

Second, the authors introduce their own meaning for "interpretability" without actually discussing it which makes it hard to follow the discussion. While usually in the literature interpretability relates to the understanding of the decisions of the black-box model and reasoning behind them, in this paper interpretability relates to the intelligibility of the robustness constraints - which of course does contribute to expert-level understandability of the model, but not much connected to the classical interpretability.

Further, the authors introduce different kind of attacks for different classes of robustness considered, but in the end everything converges to only adversarial attacks (that is only one class of robustness) without proper justification for this. Moreover, exclusively PGD attacks are used, motivated by state-of-the-art adversarial robustness checks.

The formal definitions of the robustness types are very sloppy and do not introduce all the variables used (mainly, for what x and what y the definition is considered is not included in the definition itself).

While being centered around adversarial research, the paper makes very rough mistakes with regard to it: (i) FGSM attack is using sign of the gradient, not the projected gradient (ii) the citation reference for PGD attack is absolutely wrong (it is not Gu&Rigazio paper from 2014). Moreover, adversarial training currently is much more advanced compared to FGSM training (e.g., [4]).
Lipschitz continuity of the function is defined with respect to the distance between outputs of the function (while in Definition3 in the paper the correct label is used right away). Moreover, one cannot just "assume" the value of the Lipschitz constant as it is done in the experiments.

The discussion comparing Lipschitz robustness and Standard robustness is wrong, since the difference is more that the difference of predictions for SR is fixed in \delta, while LR should become less with moving the inputs close to each other.

The introduction of Constraint Properties is given in the evaluation section which makes it unconnected to the framework definition. Moreover it is unclear how the constraints are evaluated in the experiments (possibly only due to the not proper discussion).

Finally, the authors omit comparison to the multitudes of other robustness frameworks, saying that it is out of scope of the current work, which is wrong (e.g., [5]).

minor details:
- The list of the approaches in the introduction does not correspond to the names introduced, which makes it very confusing between standard robustness and classification robustness.

[1] Mangal, Ravi, Aditya V. Nori, and Alessandro Orso. "Robustness of neural networks: A probabilistic and practical approach." 2019 IEEE/ACM 41st International Conference on Software Engineering: New Ideas and Emerging Results (ICSE-NIER). IEEE, 2019.

[2] Zheng, Stephan, et al. "Improving the robustness of deep neural networks via stability training." Proceedings of the ieee conference on computer vision and pattern recognition. 2016.

[3] Gopinath, Divya, et al. "Deepsafe: A data-driven approach for assessing robustness of neural networks." International symposium on automated technology for verification and analysis. Springer, Cham, 2018.

[4] Wong, Eric, Leslie Rice, and J. Zico Kolter. "Fast is better than free: Revisiting adversarial training." arXiv preprint arXiv:2001.03994 (2020).

[5] Gehr, Timon, et al. "Ai2: Safety and robustness certification of neural networks with abstract interpretation." 2018 IEEE Symposium on Security and Privacy (SP). IEEE, 2018.

**Summary Of The Paper:**

The authors are proposing an approach to systematization of the types of robustness of neural network. In particular, they are discussing four types of robustness: through augmentation, through adversarial training, through Lipschitz constraint training and through logical constraints training. The authors discuss the interconnections between the four classes of robustness, hypothesize which one is more general than others, propose possible measures for how easily the robustness can be violated - randomly or intentionally. In the experimental evaluation the authors are checking the attacks vulnerability for the networks with different types of robustness, confirming discussions about them.

**Summary Of The Review:**

Based on the mistakes in the text of the paper, sloppiness of the formal definitions, and unclear contribution, I recommend rejection.

---

> ### Author Response · Authors · 2021-11-22
> **Clarifications on the reviewer's doubts**
>
> > First, the restriction to only four approaches in such a vast field of research is not preferable for a general framework (just as an example, there are probabilistic robustness [1], stability training [2], targeted robustness [3]).
>
> We will investigate different approaches of robustness, thanks for the suggestions.
> For an immediate comment, adding probability to the definitions is orthogonal to the analysis we performed as all 4 definitions of robustness that we provided in the paper can be transformed into probabilistic ones.
>
> > Second, the authors introduce their own meaning for "interpretability" without actually discussing it which makes it hard to follow the discussion. While usually in the literature interpretability relates to the understanding of the decisions of the black-box model and reasoning behind them, in this paper interpretability relates to the intelligibility of the robustness constraints - which of course does contribute to expert-level understandability of the model, but not much connected to the classical interpretability.
>
> We are talking about interpretability of the constraint, while the reviewer is referring to interpretability of the model.
> While one could argue that our definition of ‘interpretability’ is connected to the classical definition because by ensuring that the network obeys understandable properties, it improves the explainability of the neural network, we agree that there can be confusion and we are happy to switch to using the term intelligibility in the paper.
>
> > The formal definitions of the robustness types are very sloppy and do not introduce all the variables used (mainly, for what x and what y the definition is considered is not included in the definition itself).
>
> To avoid repetition, we define x and y for all the robustness definitions once immediately above Definition 1: x is any input, while y is its label.
> > (i) FGSM attack is using sign of the gradient, not the projected gradient (ii) the citation reference for PGD attack is absolutely wrong (it is not Gu&Rigazio paper from 2014). Moreover, adversarial training currently is much more advanced compared to FGSM training (e.g., [4])
>
>
> Thank you for pointing these errors out, we will fix them. In particular we now cite Madry et al (https://arxiv.org/pdf/1706.06083.pdf).
>
> >  Lipschitz continuity of the function is defined with respect to the distance between outputs of the function (while in Definition3 in the paper the correct label is used right away).
>
> This is an excellent point and indeed there is a discussion in the last paragraph of Section 3 of the consequences of using either the output of the function or the correct label. For consistency with our definitions we always use the correct label, but maybe our point would be better motivated by using the more classical definitions of Lipschitz continuity.
>
> > Moreover, one cannot just "assume" the value of the Lipschitz constant as it is done in the experiments.
>
> While we understand the reviewer’s point that the true value of L needs to be experimentally determined, our experiments are simply asking the question around which proportion of points the function is Lipschitz robust for L=10. As far as we are aware this is a valid question to ask, but if it is not we would welcome an explanation of why it is not.

---

> > ### Comment · Reviewer_6o25 · 2021-11-29
> > **Answer**
> >
> > I am glad that the authors appreciate the comments and I wish luck with future versions of the paper.

---

### Official Review · Reviewer_DHPy · 2021-11-02

**Correctness:** 1
**Technical Novelty And Significance:** 2
**Empirical Novelty And Significance:** 2
**Recommendation:** 1
**Confidence:** 5

**Main Review:**

Strengths:
* This paper attempts to systematically compare different robustness definitions.

Weaknesses:
* Many arguments in the paper are too strong yet without sufficient justification and are not convincing (more in detailed comments).
* The major contribution and insights of this paper remains unclear.  Robustness can be viewed as a mathematical property, which is known. If the authors aim to make a systematic discussion, I think they need to extensively cite related works to support the insights.
* The paper is not clearly structured and is difficult for readers to get the core insights of this paper.  I suggest there to be subsections and some reorganization.

Detailed comments:
* "Definition 2 (Standard robustness)": This doesn’t really appear to be “standard” from my knowledge. I see “classification robustness” more in the literature, but this paper claims Def 2 as the “standard” without sufficient justification.
* It is unclear what “globally desirable” means. The authors put “standard robustness” as “globally desirable” but “classification robustness” as not “globally desirable”, which does not look correct to me.
* I think the “classification robustness” definition is closer to the goal of adversarial training but this paper says adversarial training is for “standard robustness”, and I think this is wrong. References are needed if the authors want to argue on that."
* “Finally we argue that, although LR is the most desirable constraint": It is not convincing why LR is the most desirable. In the rebuttal, the authors argue that “LR is strictly stronger”, but this is not convincing that being strictly stronger means more desirable. Other factors (such as the feasibility) are also important. There are similar issues with the strong argument "(S)CR should be avoided if possible”.
* Def 5~7: Literally "satisfaction", "security", "accuracy" have similar meanings, but the paper assigns different definitions to "Constraint satisfaction", "Constraint security", "Constraint accuracy", which is quite confusing .
* "This work considers four of the most prominent families of techniques": This sentence asserts the listed categories as the "most prominent" but misses lots of important works on certified adversarial robustness (including randomized smoothing and deterministic methods). To list a few examples (there are many more others):
  * Wong, Eric, and Zico Kolter. "Provable defenses against adversarial examples via the convex outer adversarial polytope." International Conference on Machine Learning. PMLR, 2018.
  * Cohen, Jeremy, Elan Rosenfeld, and Zico Kolter. "Certified adversarial robustness via randomized smoothing." International Conference on Machine Learning. PMLR, 2019.
  * Li, Bai, et al. "Certified adversarial robustness with additive noise." Advances in Neural Information Processing Systems 32 (2019): 9464-9474.
  * Salman, Hadi, et al. "Provably robust deep learning via adversarially trained smoothed classifiers." Advances in Neural Information Processing Systems 32 (2019): 11292-11303.
  * Gowal, Sven, et al. "On the effectiveness of interval bound propagation for training verifiably robust models." arXiv preprint arXiv:1810.12715 (2018).
  * Zhai, Runtian, et al. "MACER: Attack-free and Scalable Robust Training via Maximizing Certified Radius." International Conference on Learning Representations. 2019.
  * Zhang, Huan, et al. "Towards stable and efficient training of verifiably robust neural networks." ICLR 2021.
  * Zhang, Bohang, et al. "Towards Certifying L-infinity Robustness using Neural Networks with L-inf-dist Neurons." International Conference on Machine Learning. PMLR, 2021.
  * Shi, Zhouxing, et al. "Fast Certified Robust Training with Short Warmup." Advances in Neural Information Processing Systems 34 (2021).


**Summary Of The Paper:**

This paper attempts to compare different robustness definitions. The paper discusses the relations between various definitions and conducts experiments to show the relations.

**Summary Of The Review:**

There are too many strong arguments or definitions that do not have sufficient justification and are not convincing or reasonable. And this paper is not quite clear and not well-structured. Author response does not address my concerns and the authors did not show up to answer my follow-up questions.

---

> ### Author Response · Authors · 2021-11-22
> **Clarifications on the reviewer's doubts**
>
> >  "This work considers four of the most prominent families of techniques": This sentence asserts the listed categories as the "most prominent" but misses lots of important works on certified adversarial robustness (including randomized smoothing and deterministic methods).
>
> As mentioned at the start of the paragraph, here we are referring to training techniques, not formal verification techniques. To clarify this we will add the word “training” to the sentence.
>
> > "Definition 2 (Standard robustness)": Why is this definition "standard"? Is it a widely adopted "standard" definition? References for justification are needed.
>
> We call it “standard robustness” because this definition is the closest to capture the natural language definition: a small change in the input should result only in a small change in the output. We will add some citations from the mathematical robustness literature.
>
> > Table 1: I don't understand why Standard robustness is "Globally desirable". According to the definition, it looks local around $\hat{x}$.
>
> Standard robustness indeed defines a local property around a single point in the input space but, as discussed under the “dataset assumption” heading in section 3 and unlike classification robustness, we want it to hold for all such points in the input space, hence why we say that it is “globally desirable”.
>
> > Still Table 1: Why do you say classification robustness doesn't have a loss function? I think 0-1 loss is also a loss function.
>
> We mean that Classification robustness does not have a loss function suitable for training, i.e. differentiable. We will add table notes to clarify.
>
> > Table 1 again: Why do you say adversarial training is for "standard robustness" rather than "classification robustness"? I think it's for classification robustness. References are needed to justify that adversarial training has a goal on the "standard robustness" you defined.
>
> We say that adversarial training is for ‘Standard robustness’ because the augmented loss function tries to minimize the distance between f($\hat{x}$), the output of the neural network and y, the training label, in the same way that standard robustness imposes a maximum distance δ between them. On the other hand, classification robustness uses no such distance function.
>
> > "Finally we argue that, although LR is the most desirable constraint": How do you determine it's the "most" desirable?
>
> This is an excellent point, our arguments are scattered throughout the paper. We will summarize them for the reader at this point in the paper.
> As discussed in section 3 under the “Standard and Lipschitz robustness” heading, LR is strictly stronger than SR, and as discussed in section 2 under the “Training with logical constraints” heading CR does not have a differentiable loss function whilst SCR does not accurately capture the desired semantics.
>
> > The last paragraph in Sec. 3 is difficult to understand and is the argument "(S)CR should be avoided if possible" is too strong without sufficient justification.
>
> The justification for this are discussed in the previous point.
>
> > Def 5~7: I don't understand what's the difference between words "satisfaction", "security", "accuracy" in the context of evaluation, but the paper assigns different definitions to "Constraint satisfaction", "Constraint security", "Constraint accuracy", which is quite confusing.
>
> I am afraid we don’t quite understand the reviewer’s question. The different definitions explain the differences. Nonetheless we will try and make the explanation more clear.

---

> ### Comment · Reviewer_DHPy · 2021-12-04
> **Post-rebuttal update**
>
> After the rebuttal phase, I have re-checked the paper and edited my review. Unfortunately, author response does not address my concerns and the authors did not show up to answer my follow-up questions. I would maintain my initial recommendation.

---

### Author Response · Authors · 2021-11-22
**Thanks to the reviewers**

We thank all the reviewers for the feedback.

---

### Decision · Program_Chairs · 2022-01-20

**Decision:**

Reject

**Comment:**

The paper studies and compares different notions of robustness. However, reviewers found there are many unjustified claims in the analysis, and the paper does not provide novel findings nor useful approaches.